# How well do models of visual cortex generalize to out of distribution samples?

Yifei Ren[1], Pouya Bashivan [1,2,3] *

**1** Department of Computer Science, McGill University, Montreal, Canada, **2** Department of Computer Physiology, McGill University, Montreal, Canada, **3** Mila, Université de Montréal, Montreal, Canada

* pouya.bashivan@mcgill.ca

**Data Availability Statement:** The data analyzed in this work is publicly available online at https:// github.com/dicarlolab/npc. Code for replicating the main paper figures is available at https://github.

## Abstract

Unit activity in particular deep neural networks (DNNs) are remarkably similar to the neuronal population responses to static images along the primate ventral visual cortex. Linear combinations of DNN unit activities are widely used to build predictive models of neuronal activity in the visual cortex. Nevertheless, prediction performance in these models is often investigated on stimulus sets consisting of everyday objects under naturalistic settings. Recent work has revealed a generalization gap in how predicting neuronal responses to synthetically generated out-of-distribution (OOD) stimuli. Here, we investigated how the recent progress in improving DNNs' object recognition generalization, as well as various DNN design choices such as architecture, learning algorithm, and datasets have impacted the generalization gap in neural predictivity. We came to a surprising conclusion that the performance on none of the common computer vision OOD object recognition benchmarks is predictive of OOD neural predictivity performance. Furthermore, we found that adversarially robust models often yield substantially higher generalization in neural predictivity, although the degree of robustness itself was not predictive of neural predictivity score. These results suggest that improving object recognition behavior on current benchmarks alone may not lead to more general models of neurons in the primate ventral visual cortex.

## Author summary

Inspired by the neural circuits of the brain, deep neural networks (DNN) have been steadily improving in their ability to perform foundational visual tasks such as object recognition. Whereas, early models struggled with generalization to abstract visual domains such as line drawings and cartoons, recent advancement have approached near-human recognition capabilities. Moreover, the unit activity in these networks exhibit strong similarities with the activity of single-unit recordings along the primate ventral visual cortex. This capability of DNNs has provided visual neuroscientists with precise models for exploring the neural underpinnings of object recognition. Our research probes whether enhancements in neural networks' recognition of out-of-distribution objects correlate with improved predictability of brain activity in the visual cortex of monkeys to synthetic stimuli. We found that the out of distribution object recognition performance on natural

com/BashivanLab/
NeuralPredictionGeneralizationGap.

**Funding:** This research was supported by the
Healthy-Brains-Healthy-Lives startup supplement
grant and the NSERC Discovery grant RGPIN-
2021-03035. P.B. was supported by FRQ-S
Research Scholars Junior 1 grant 310924, and the
William Dawson Scholar award. The funders had
no role in study design, data collection and
analysis, decision to publish, or preparation of the
manuscript.

**Competing interests:** The authors have declare
that no competing interests exist.

image datasets is not a reliable measure of neural predictivity. However, DNN models that
were trained to be more resilient to adversarially generated noise patterns as well as DNN
ensembles, consistently yielded better generalization in neural predictivity. Altogether,
our results suggest that improving object recognition behaviour on current benchmarks
alone may not lead to more general models of neurons in the primate ventral visual
cortex.

## Introduction

*Generalization* is a hallmark of biological intelligence. Animals in general and primates in par-
ticular possess a spectacular ability to *transfer* or *generalize* their knowledge to novel situations.
For example in vision, humans and non-human primates can effortlessly recognize objects
and animals from their natural photos, drawings, line drawings, and even can distinguish
between objects from their cloth-covered 3D models [1].

Formally, *generalization* is often defined as the ability to transfer acquired knowledge to
new problems. In the domain of sensory perception, generalization is regarded as spontaneous
transfer of knowledge from a source domain to a target domain. Although, the very definition
of what constitutes a domain remains elusive and often subjectively defined by human experts.
As a consequence, we currently lack reliable metrics that quantify the degree of similarity
between two visual domains without any humans in the loop. In the absence of any agreed
upon quantitative metrics for measuring the difference across domains, the human judgement
of domain dissimilarity is taken as the ground truth. Examples of such domains in vision
include photos of natural objects in the wild, drawings, cartoons, and computer generated syn-
thetic objects.

In recent years, artificial intelligence powered by deep artificial neural networks has made
significant progress in foundational visual tasks such as object recognition [2–4], object detec-
tion [5–7], and even scene generation from natural language descriptions [8–10]. These
advances suggest that the units in such deep neural networks may learn domain-general visual
features similar to those that are hypothetically represented by the neurons in the animal
brains. However, later work discovered that these networks struggle to transfer to input
domains that remain unseen during training [11]. To facilitate progress in building models of
object recognition, several benchmarks were developed, aiming at assessing the ability of rec-
ognition models across different domains of objects. In particular, these benchmarks targeted
more difficult natural photos including objects in unfamiliar contexts and forms, drawings
and sketches, and natural and artificial perturbations to natural photos [12–16].

Despite their apparent flaws, DNNs have also been widely influential on other domains of
science including in neuroscience [17, 18]. In the vision domain, there is mounting evidence
demonstrating an unprecedented similarity between the internal activity of these models and
brain activity measurements across modalities (fMRI, ECOG, EEG, MEG, and electrophysio-
logical recordings) [19–21]. More remarkably, the same class of models have been utilized to
not only predict the neuronal activity but also to induce desired response patterns in single
neurons as well as small populations [22–24]. Despite these remarkable feats, DNNs' predictive
power is much weaker when they are tested on synthetically generated stimuli [23, 24]—signal-
ing an apparent limitation in generalization as models of neurons and neuronal computations
in the brain. Given that a fundamental goal of computational neuroscience is to replicate the
brain function using mathematical and statistical equations, we desire models that can accu-
rately capture the function of elements in the nervous system across all possible conditions

imagined. In the context of predictive models of brain, this means that predictions from such models should ideally generalize well to input distributions that were not used in constructing the models.

Considering the more recent advances of DNNs in closing the gap in cross-domain invariant object recognition, it is natural to ask: *Do object recognition models with better generalization capacity also constitute more accurate and more general models of neurons in the brain?* In this work, we set out to seek an answer to this question by assessing and comparing a large number of DNN models of visual object recognition in their ability to predict the neural activity across various stimulus domains. We found that the models' performance on none of the common object recognition benchmarks was a strong indicator of the generalization performance of the neuronal models derived from their unit activations. While the same finding was extended to adversarially robust models, we found that the neuronal models stemming from most robust models have better generalization properties in neuronal predictivity compared to their non-robust counterparts. Furthermore, by taking a reductionist approach, we sought to identify individual factors that control the generalization in neuronal models. Our experiments showed that network architectural properties such as depth and width as well as a specific unsupervised learning algorithm called Momentum Contrast [25] can substantially boost the neuronal models' generalization performance. Finally, we examined the neuronal prediction consistency across many neuronal models stemming from different neural network architectures and demonstrated that ensemble predictive models constructed from combining predictions made by multiple neuronal models could leverage the differences across these models to build more accurate predictive models of neurons.

## Results

Predictive modeling is one of the prominent approaches for evaluating representational similarity between neural network models and the brain [19, 21, 22, 24, 26, 27]. In this framework, the similarity between a model and the brain is measured in terms of the accuracy of the image-level neuronal response predictions, made from unit activations in a particular neural network model. This procedure is often performed on a dataset of visual stimuli selected from the same semantic domain (e.g. photos of natural objects or Gabor filters of different frequencies and orientations). The stimuli are cross-validated to avoid reporting results from an overfitted model. While this approach has been largely informative with regard to identifying the models that more closely replicate the neural representations in animals' visual cortex, *the extent to which such models can generalize their predictions to stimuli from other semantic domains, is currently unknown*. Knowledge of such generalization properties is important because the ultimate goal of these modeling exercises is to attain a model of the neurons in the animal brain that mimics their response variations under *any* experimental circumstances and specifically under those unseen during model construction.

To investigate this question, we evaluated a large number of visual neural network models (38 models, see section Model selection for a full list) on a number of in-distribution and out-of-distribution object recognition benchmarks, while also assessing their representational similarity with neuronal response properties in the macaque visual cortex to natural and synthetically generated stimuli (Fig 1). In section Gap in neural predictivity across domains, we examine the generalization gap in neural predictivity and establish its widespread existence across models of ventral visual stream. In section Assumptions on brain-model correspondence and their effect on generalization, we scrutinize the validity of the assumptions underlying two prominent approaches to evaluate neural predictivity in neural network models. Section Is object recognition generalization ability a reliable indicator of neural predictivity?

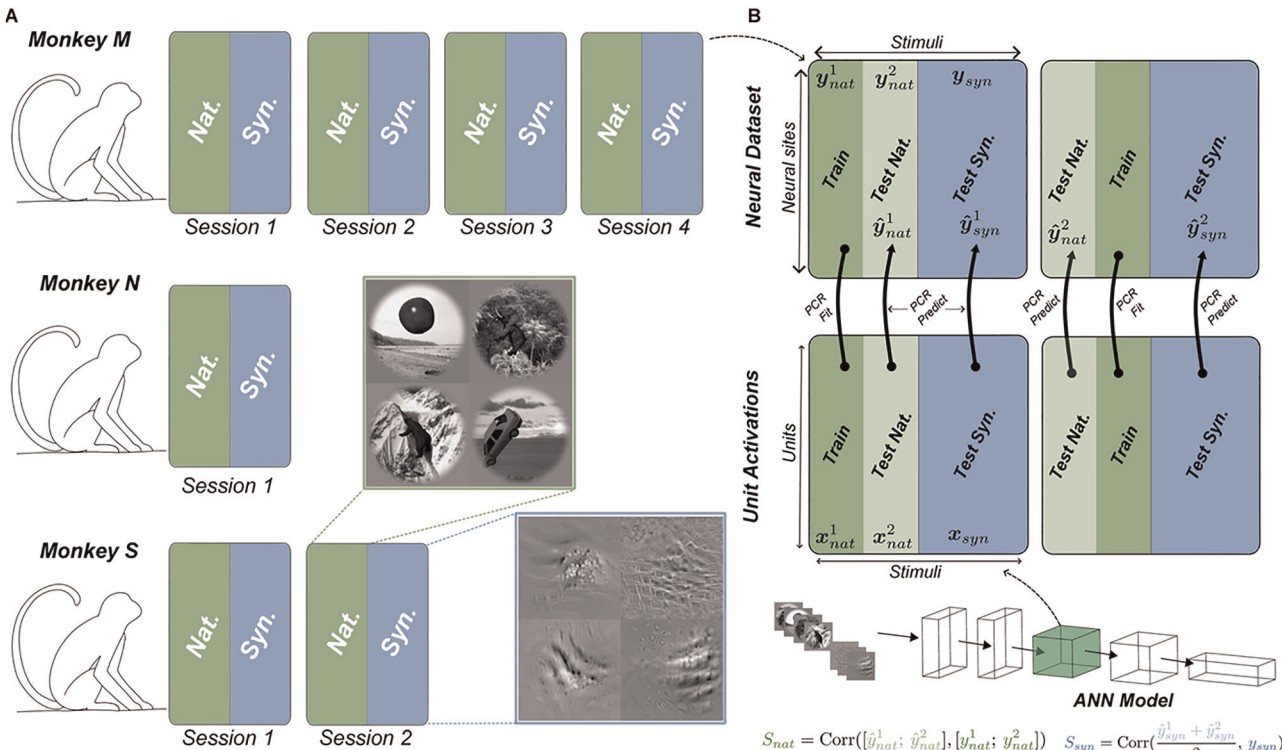

**Fig 1. Neural predictivity as a measure of model-brain similarity.** A: We used electrophysiological recordings from area V4 in three macaque monkeys. Each animal's data was collected across one or several recording sessions (1–4 sessions). Data corresponding to each session included measured neural responses (firing rate) to a set of naturalistic (i.e. Nat.) as well as a set of synthetically generated (i.e. Syn.) images. B: Illustration of the cross-validation procedure (2-fold) using principal component regression (PCR) for computing the neural predictivity score for naturalistic and synthetic stimuli. The natural dataset (top) is split randomly into two partitions (i.e. folds) along the stimulus dimension. For each fold, one partition is used to fit the PCR model towards predicting the neural data from neural network model unit activations in a given layer. The resulting PCR model is used to predict the responses to both the natural stimuli in the test-partition and the synthetic stimuli. The similarity scores for each stimulus domain ($S_{nat}$ and $S_{syn}$) is computed by combining the predictions across the two folds.

explores the relationship between generalization in object recognition performance and neural predictivity. Furthermore, we analyze the impact of design choices such as network architecture and learning objective on generalization gap in neural predictivity in section Modeling design choices shape generalization in neuronal models. Lastly, in section Do different models predict the neuronal responses the same way?, we assess the extent of overlap among different models of ventral visual stream and proposed a way to construct more accurate predictive models by construction of model ensembles.

## Gap in neural predictivity across domains

We first examined how good typical neuronal models (i.e. predictive models of neuronal responses) transfer from one stimulus domain to another. For this, we considered a dataset of electrophysiological recordings from macaque area V4 to stimuli from two readily distinguishable visual domains from a recent study [24]. The first visual domain which we refer to as naturalistic domain or in-distribution (ID), contains stimuli where each one consists of the rendering of a natural 3D object instantiated at a random location, size, and pose and overlaid on a random natural image background. The second domain which we refer to as the synthetic

domain or out-of-distribution (OOD), contains synthetically generated pixel patterns that were produced using an optimization procedure described in [24]. These images contained abstract patterns and shapes but were detached from any nameable objects (Fig 1A).

We used a standard procedure (cross-validated Principal Component Regression, see Methods and Fig 1B), to measure the prediction accuracy for each model and considered a pool of 38 artificial neural network models in our analyses (A detailed model list is described in section Model selection). Each neural network model consisted of multiple layers of computation with varying number of units in each layer. In each model, we assessed the accuracy of predictions made from unit activations in each model's layer separately. We considered the unit activations in each layer of neural network as a possible representation (i.e. feature) space that could potentially be linearly combined to explain the response patterns in neuronal sites. A regression model was fitted to predict the in-distribution neural data from the network unit activations in a particular layer. We refer to the collection of the neural network layers that were used to produce the unit activations together with the additional regression model as the "neuronal model". Each neuronal model was evaluated through its "neural predictivity score" (see Methods), which was measured via the accuracy of its predictions on held out naturalistic images and all stimuli from the synthetic domain.

Overall, neuronal models constructed from different neural network architectures predicted the image-level neuronal responses to stimuli from both domains above chance and training the network parameters improved neural predictivity in both domains (S6 Fig). However, for many neuronal sites, the in-distribution predictions were much more accurate than those for the out-of-distribution samples (Fig 2D), and models with higher in-distribution neural predictivity tended to have higher out-of-distribution scores as well (S2 Fig).

To better understand the source of this gap, we examined the neural predictivity scores on two domains against each other (Fig 2A—per neuron scatter plot of in-distribution vs. out-of-distribution neural predictivity). The neural dataset contained a total of 232 neurons in total across all sessions and animals. In our analyses, we only considered neurons with an internal consistency value of greater than 0.7 which resulted in having 145 neurons from OOD data sessions and 224 neurons from ID data sessions.

We found that 85% of the neuronal sites were predicted well on the ID domain (correlation >0.6) and the prediction score for none of the sites was below 0.4. In contrast, only 38% of the neurons were predicted well on the OOD domain (correlation >0.6) while 40% of the neurons were not well predicted in that domain (correlation <0.4) (Fig 2B). The result of this analysis suggested that the primary reason for the gap in cross-domain neural prediction is due to a large set of neurons that are predicted with high accuracy in the naturalistic domain but not in the synthetic domain. This cross-domain gap in neural predictivity score existed when predicting the neuronal measurements in at least some neuronal sites for all three animals and across different sessions (Fig 2C) and also across different models of ventral visual stream (Fig 2D). Analysis of data from one monkey (Monkey-S) showed systematically lower generalization gap which may be attributed to 1) the higher eccentricity of neurons recorded in this monkey compared to others or; 2) the fact that many of the synthetically generated images for this monkey covered a smaller part of the images compared to those that were generated for other monkeys. The images covering larger parts of the field could have potentially led to invoking more complex population responses that were more difficult to be captured by the DNN models tested in this work.

It is possible that the features corresponding to the neurons in the brain exist in the neural network model; however, the right combination of these units may not be found because of limitation in the diversity of our samples within the naturalistic domain. To investigate this hypothesis, we also fitted a regression model on data from the synthetic domain and tested

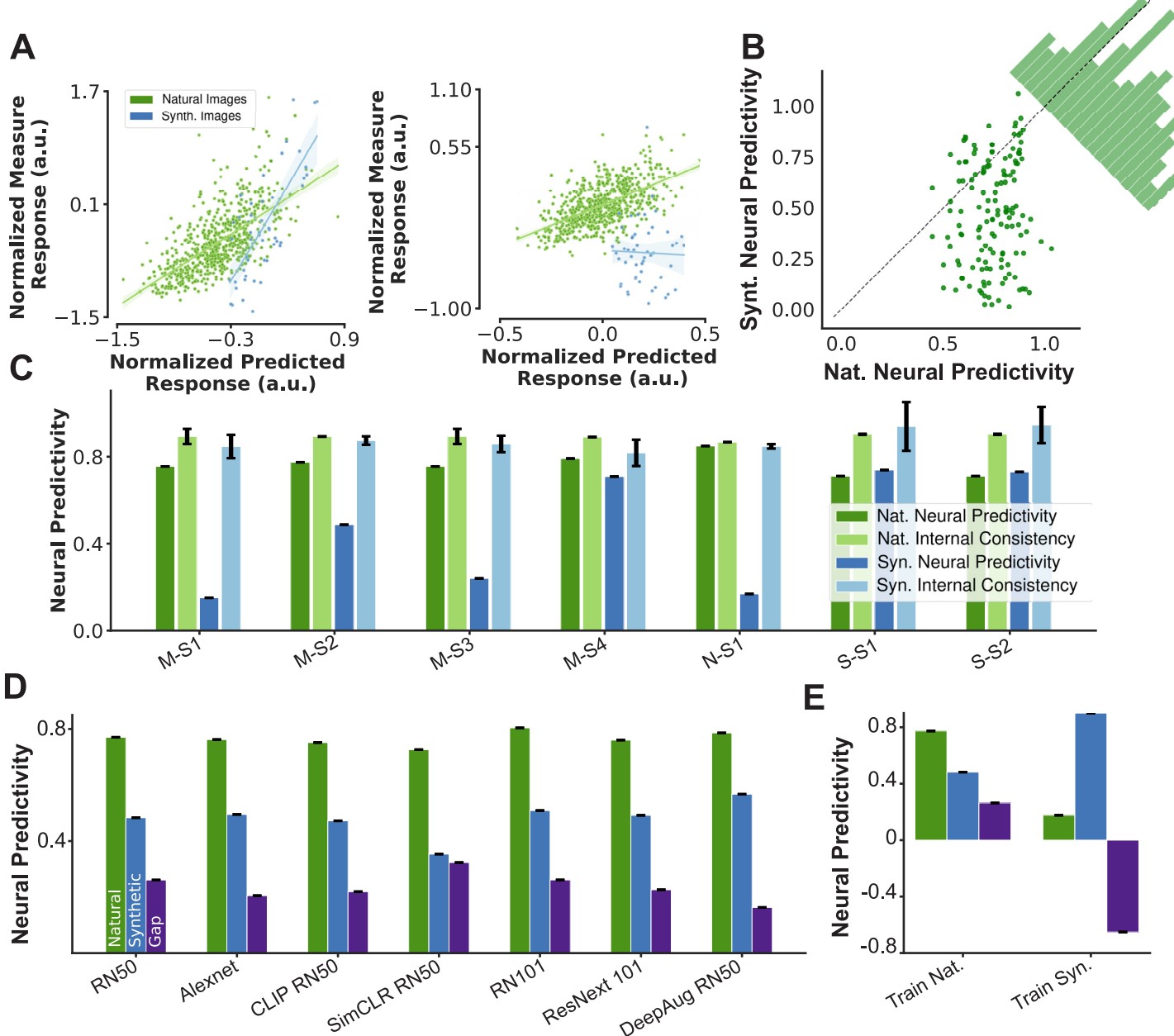

**Fig 2. Gap in neural prediction.** A: Scatter plot of normalized measured responses and their corresponding predicted values from a layer in ResNet50 model for two sample neuronal sites. The neuronal model's generalization capability is highly variable across neurons. Left and right plot show two examples neuron with high and low generalization respectively. B: Scatter plot of Nat. and Syn. predictivity scores for a neuronal model based on ResNet50 unit activations for all neuronal sites with high internal consistency (larger than 0.7). The corner histogram shows the distribution of the difference between in- and out-of-distribution predictivity scores across neuronal sites; C: Neural predictivity score of the ResNet50 neuronal model and internal consistency of the neural data in naturalistic and synthetic domains for neural data collected from different animals (M, N, and S) and different recording sessions (S1–4); D: Bar plot of Nat. and Syn. predictivity scores as well as the neural predictivity gap for 7 different neural network models; E: Comparison of Nat. and Syn. predictivity scores for ResNet50 model when the regression model was fitted on naturalistic data (left) and synthetic data (right). The regression model fitted to the synthetic domain shows worse generalization to the naturalistic domain. All error bars denote the variance across 5 repetitions of each analysis.

that model's predictions on the naturalistic and synthetic domains (Fig 2E). This model achieved a higher prediction accuracy on the synthetic domain but yielded almost no transfer to the naturalistic domain. This result demonstrated that the unit activations in the neural network model could predict the neuronal responses rather well within each domain, but failed to transfer most of their prediction power across domains.

## Assumptions on brain-model correspondence and their effect on generalization

Each neural network model of ventral visual cortex typically consists of multiple stages of processing known as layers. The number of layers in modern deep neural networks often substantially exceeds the number of anatomical areas identified along the Ventral Visual Stream (VVS). *Given the apparent lack of one-to-one mapping between layers in the neural network models and brain areas, where should we look for model units with similar response profiles?*

The standard procedure for assessing representational similarity between a model and a particular brain area commonly involves selecting all units within a layer of the neural network as candidate predictors and using cross-validation to fit a simple mapping function (a variation of a regression function in most cases) to the observed neuronal responses. Importantly, the implicit assumption underlying this procedure is a one-to-one mapping between a layer in the neural network and a particular brain area. Despite its wide usage in the literature, such strong assumption may be over-restrictive in practice, specially when considering the typical number of layers in neural network models that are substantially higher than the known number of areas along the VVS.

A natural alternative to this assumption is to consider possible correspondence between one brain area and multiple model layers. In this view, neurons in one brain area could correspond to an aggregation of units in multiple neural network model layers. In its most general form and without any prior, it is numerically cumbersome to test all possible layer combinations as candidate predictor sets for representing each brain area, or to consider all layers simultaneously in establishing the mapping between the model and the brain. However, one feasible alternative to that is to assume that each individual neuron can be approximated by units within a single layer of the neural network model but also that not all measured neuronal sites from the same brain area may necessarily correspond to the same model layer.

We investigated which one of the two approaches best explains the relationship between neurons and units arranged across layers of neural network models. For this, we first examined the neural prediction generalization properties of neuronal models constructed from different layers of an ANN model (ResNet50) for each individual neuronal site. We constructed a neuronal model for each neuronal site in our dataset from each layer of the ResNet50 model and evaluated each model's neural prediction accuracy on naturalistic and synthetic stimuli. We found that 1) many of the model layers had similar levels of in-domain neural predictivity (Fig 3A). Indeed, for many neuronal sites, the neural predictivity score for the first model layer was almost as high as that obtained for the best model layer; 2) most neuronal sites were best predicted from unit activations in two or three of the model's layers (Fig 3D); 3) the best predictive layer for each neural site varied across stimulus set (naturalistic vs. synthetic). For a large number of neurons, models constructed from units in later layers of the network had substantially higher generalization capacity to synthetic stimuli while having more or less similar generalization on the naturalistic domain (Fig 3B). In ResNet50, we found that the layer with the highest generalization to out-of-distribution samples was on average 1.29 layers later than that corresponding to in-distribution samples (Fig 3C).

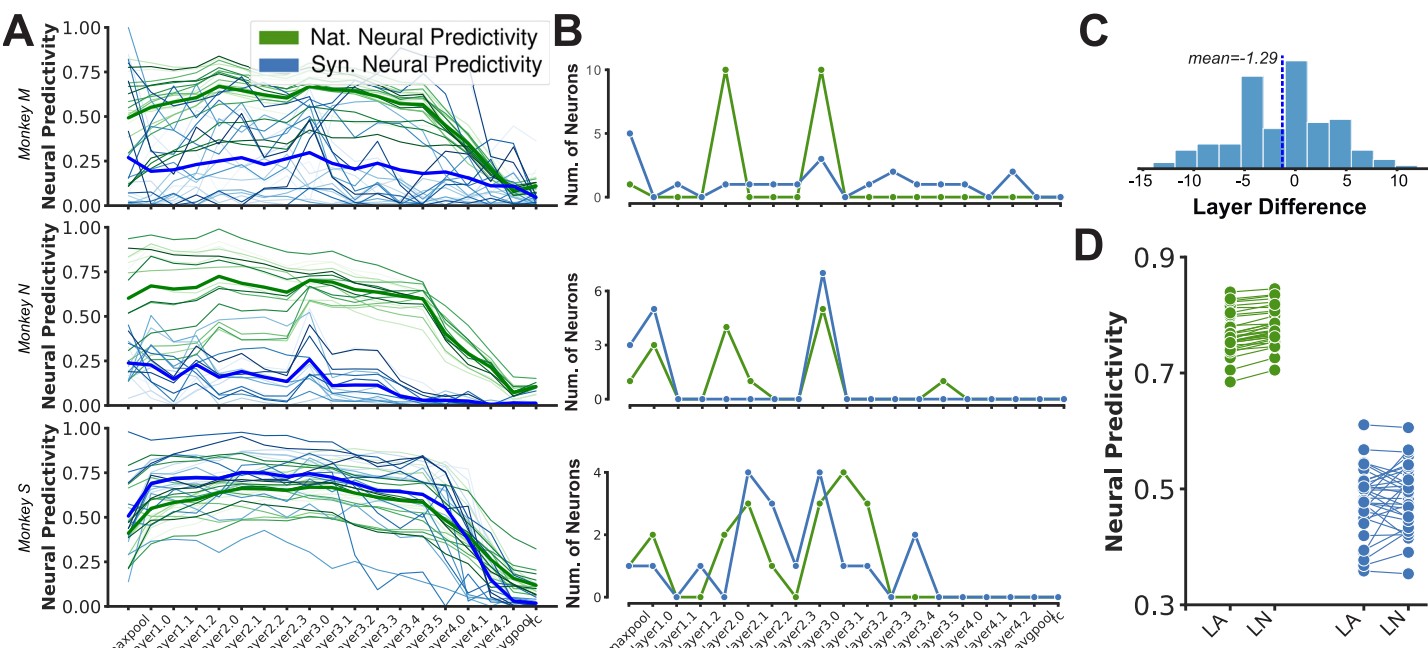

**Fig 3. Assumptions on brain-model correspondence affect generalization in neural predictivity.** A: Neural predictivity scores from unit activity in each layer of ResNet50 architecture for individual neuronal sites recorded during example sessions from different animal subjects. From top, rows correspond to M-S1, N-S1 and S-S1. Colors correspond to the neural predictivity score on natural (green) and synthetic (blue) domains. Different shades correspond to different neuronal site in the same animal. Bold lines correspond to the average predictivity score in each domain across all neuronal sites within that animal's session; B: Number of neurons with highest neural predictivity in a given layer corresponding to the same subplot in a. Colors are the same as in a; C: Distribution of the difference between the layer number in ResNet50 neural network where each neuronal site is best predicted in-distribution and out-of-distribution. The difference is calculated as the best ID layer number—the best OOD layer number. The distribution spans a wide range but has a slightly negative mean (-1.29); D: Comparison of neural predictivity scores on natural and synthetic domains when a brain-model correspondence follows a Layer-Area (LA) or Layer-Neuron (LN) mapping assumption.

Next, we investigated the effect of these assumptions on model-brain correspondence on neural prediction generalization. For this, we constructed predictive models of the neuronal population in two ways: a) **Layer-Area mapping:** following the common approach in the literature [19, 24, 28] by taking each neuron's model from the layer with the highest in-distribution cross-validated median prediction accuracy across the neuronal population or; b) **Layer-Neuron mapping:** taking each neuron's model from the layer with the highest in-distribution cross-validated prediction accuracy for that neuron (see Methods). We found that a more flexible assumption on the model-brain correspondence (Layer-Neuron mapping) leads to higher generalization on in-distribution samples and that this improvement was consistent across different model architectures (0.78 vs. 0.768; paired t-test: p <0.0001) (Fig 3D). On out-of-distribution samples, the Layer-Neuron mapping assumption does not significantly improve the predictivity scores across all models (0.487 vs. 0.479; paired t-test: $p = 0.28$). While the difference between the two mapping methods appear small when judged by the median statistic, they are occasionally very large and positive for individual neural sites when the distribution of these differences are examined (up to 0.4 increase in predictivity on OOD; S7 Fig). Overall, these results supported our initial hypothesis that the assumptions behind layer-area approach are too restrictive and that layer-neuron mapping may provide a more unbiased view of the similarity between a neural network model and a given neural dataset.

## Is object recognition generalization ability a reliable indicator of neural predictivity?

There is substantial evidence that the primate ventral visual stream underlies their ability to recognize objects [29, 30]. Modeling experiments have also shown that the internal activity of neural network models trained to perform visual object recognition tasks, are highly similar to the neuronal responses measured from the ventral visual cortex [19, 31]. Furthermore, several prior works have demonstrated a strong relationship between measures of object recognition behavior and neural similarity [19, 28]. Based on these previously observed similarity patterns, we reasoned that models with better generalization capabilities in visual object recognition may also produce better generalization properties in their corresponding neural predictors. In other words, we may reasonably expect that because these models are capable of transferring their ability to recognize objects across different semantic domains, the units in these models may establish a closer relationship with the neurons in the brain.

To investigate this hypothesis, we assessed the performance of each neural network model (33 models including the original 38 except those without the last linear classification layer; InsDis ResNet50, Local Aggregation ResNet50, Place365 ResNet50, MoCo ResNet18, MoCo ResNet101) on 5 commonly used object recognition benchmarks from machine learning literature [12–15, 32]. We then examined the relationship between each model's performance on these benchmarks with the neural predictivity scores within the naturalistic and synthetic domains. Each of the adopted benchmarks uniquely probed the neural network models on their object recognition generalization behavior that included natural photos of naturally occurring objects, object renditions, natural objects in difficult settings, drawings, and natural corruptions (Fig 4A). The recognition performance of typical neural network models varied greatly on each of these benchmarks (Fig 4B).

**Object recognition accuracy on the ImageNet dataset.** As a first measure, we considered the classification accuracy on the widely used ImageNet dataset to assess the models' object recognition performance under common natural settings. The parameters of most existing models of ventral visual cortex (including most of those considered in the present study) are tuned on images taken from the train set of this dataset. We computed each model's object recognition accuracy on examples from the ImageNet validation set (not used for parameter tuning of any of the models). We then computed the Spearman correlation between ImageNet object recognition accuracy and the neural predictivity score across models for each of the two domains (i.e. natural and synthetic). The ImageNet object recognition accuracy showed a trend of positive correlation with the neural prediction accuracy on synthetic images but it did not exist for the neural prediction measures on natural images. Furthermore, neither of these relationships were statistically significant (p = 0.46 and p = 0.07 for natural and synthetic domains respectively; Fig 4C).

**Robustness to out-of-distribution Samples.** While many existing neural network models can perform object recognition at a level similar or better than human subjects on the ImageNet dataset [33], it was soon discovered that most models' object recognition accuracy substantially declines when evaluations are carried out on images that are collected from alternative semantic domains or those with natural or artificial alterations. Several benchmarks were proposed to quantify the progress towards more general object recognition models. Since then, there has been a remarkable progress in improving invariant object recognition in neural network models and consequently, many recent neural network models can now recognize objects even when they are observed from unfamiliar views or in abstract forms like in line drawings [3].

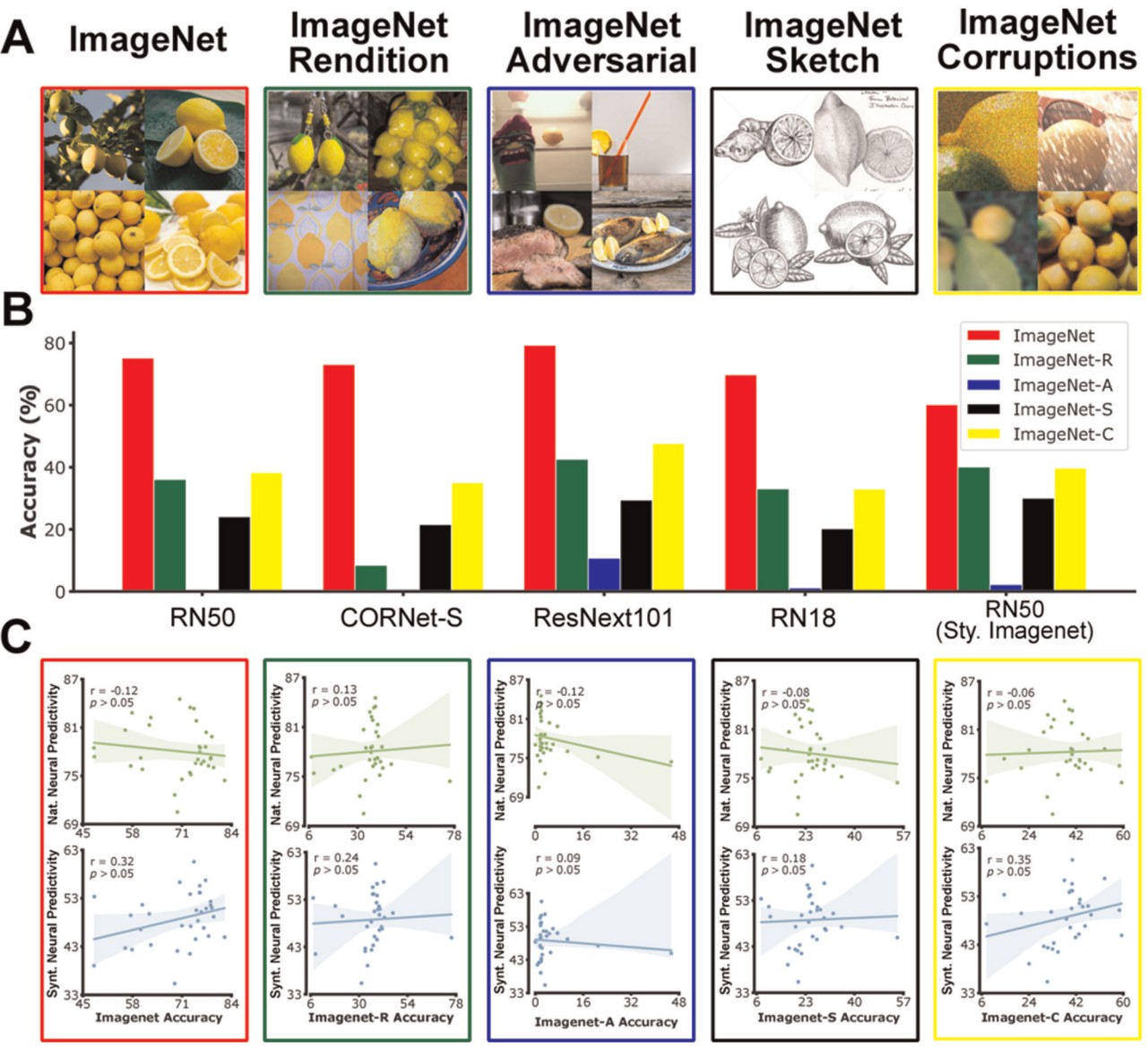

**Fig 4. Object recognition performance and neural predictivity.** A: Example images from five object recognition benchmarks. Due to copyright restriction, sample ImageNet images are publicly available images that are qualitatively similar to those within that dataset. B: Object recognition performance of five example neural network models on different benchmarks. Object recognition performance on out-of-distribution benchmarks are significantly lower than those obtained on the ImageNet validation set. C: Measures of object recognition performance do not correlate with neural predictivity scores on natural or synthetic domains.

We further evaluated the models' object recognition performance on four out-of-distribution recognition datasets including a) ImageNet-Rendition [12] that contains naturally occurring examples which are renditions (e.g., paintings, embroidery, etc.) of 200 object classes in ImageNet classes, but with textures and local image statistics unlike those of ImageNet images; b) ImageNet-Adversarial that contains a set of natural photos which are selected through an adversarial filtration process that only includes images that cannot be solved by simple spurious cues [13]; c) ImageNet-Sketch [14] that contains sketch-like images for each of ImageNet

classes; d) ImageNet-Corruptions [15] that contains natural images from ImageNet validation set that are corrupted with 16 different types of natural corruption procedures including various types of noise, weather-effects, and blurs (see Methods for the full list).

To measure the model's OOD generalization performance, we computed the average classification accuracy of each model on each of the four OOD datasets. For each dataset, we then correlated the object recognition accuracy with neural predictivity scores across models for each of the two domains (Fig 4C). We found that the performance on none of the OOD recognition benchmarks was significantly correlated with neural prediction score in natural or synthetic domains. In other words, in the context of these benchmarks, the improvements on the OOD object recognition does not translate into improvements in neural prediction accuracy in natural or synthetic domains. OOD object recognition gap was also not correlated with neural prediction generalization gap (S1 Fig), which indicates that improvements in OOD object recognition does not seem to be closing the gap in neural predictivity.

**Robustness to adversarial perturbations.** Another metric for evaluating generalization in neural network models is adversarial robustness [34, 35]. Robustness metrics use optimization tools to find extremely small image perturbations that can drastically change a model's generated output. The perturbations are often constrained such that the resulting image would not differ much from the original image. In most cases, the resulting "adversarial examples" remain indistinguishable by humans from the original ones. It has been proposed that the adversarial examples could be considered as samples from an input distribution which by definition is designed to produce incorrect object category predictions in the target model [36, 37]. While the object recognition performance of many models significantly declines under these perturbations, various approaches have been proposed to reduce the recognition gap under such adversarial settings [38–42].

Here, we investigated whether the degree of each model's robustness to adversarial perturbations is predictive of their corresponding neural predictivity. To evaluate each model's robustness, we used a procedure known as Projected Gradient Descent (a form of "adversarial attack"), to find small perturbations to the images which maximally reduce the model's confidence in its prediction of the correct object class associated with an image. We evaluated a neural network's robustness by measuring the accuracy of predictions on the resulting adversarial examples, and computed the correlation between these values with neural predictivity across models.

As shown in Fig 5A (top), models' robustness to adversarial perturbations was significantly correlated with their neural predictivity on naturalistic domain but not on the synthetic domain (Fig 5A (bottom)). Inspecting robustness measures with specific norms and $\varepsilon$ revealed that the observed significant correlation in natural domain is primarily driven by the correlation between measures of robustness computed at small $\varepsilon$ values (S5 Fig). Despite the lack of correlation in the synthetic domain, we noted that most models that were trained specifically to improve their robustness to adversarial perturbations, had higher neural predictivity scores compared to the non-robust variation of the same architecture (ResNet50; Fig 5B). We also found that the degree of neural prediction accuracy varied systematically as a function of the magnitude of adversarial perturbations to which the models were exposed during training, and the pattern formed an inverse U-shape (Fig 5C) where the highest neural predictivity was obtained at intermediate values of perturbation magnitude ($\varepsilon$).

Moreover, we investigated whether the increase in neural predictivity of adversarially robust models was caused by improvements in specific subset of neuronal sites or uniformly across all neuronal sites. To do this, we compared the neural predictivity between a robust ($\varepsilon_{L_2} = 0.25$)) and non-robust ResNet50 model across individual neuronal sites. We found that

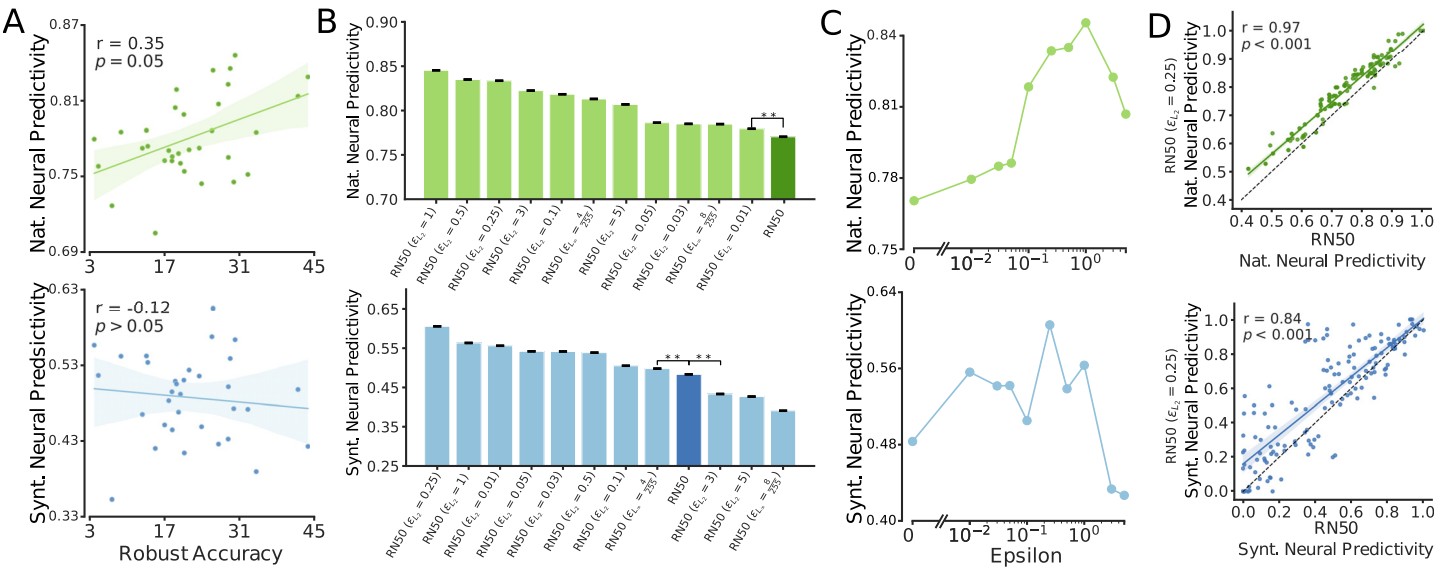

**Fig 5. Robustness to adversarial perturbation improves neural predictivity.** A: Scatter plot of neural predictivity on natural (top) and synthetic (bottom) domains as a function of robust accuracy (i.e. average of accuracy under several adversarial perturbations); B: Comparison of neural prediction score on naturalistic (top) and synthetic (bottom) stimuli across a range of models optimized for improved robustness. All error bars denote the variance across 5 repetitions of each analysis; C: Effect of train-time perturbation magnitude on neural predictivity. Here, we only included the robust models with $L_2$ norm and similar attack settings and excluded 5 models that use different attack norm, number of steps, and step sizes; D: Scatter plot of neural predictivity scores for ResNet50 and ResNet50 ($\epsilon_{L_2} = 0.25$) models. Each dot corresponds to an individual neuronal site.

in the natural domain, the improvements to neural predictivity existed almost uniformly across all neuronal sites. However, in the synthetic domain, while improvements still existed in most neuronal sites, they were more variable and particularly large for a subset of sites (Fig 5D).

## Modeling design choices shape generalization in neuronal models

Results in the previous section demonstrated an apparent absence of relationship between generalization in object recognition performance and neural predictivity score. However, neural predictivity and its generalization may be attributed to other characteristics of these networks that are not exactly reflected onto their recognition behavior. We asked whether the variations in neural predictivity and generalization to out-of-distribution samples may be the product of the network architecture, learning objective, or the dataset used for training. To answer this question, we next investigated how individual design factors determining the architecture and connection weights in each model, affect their neural prediction generalization.

**Network architecture.** A number of recent work has studied the scaling properties of neural networks and how performance in many neural network models can be systematically boosted with their increased depth and width [43, 44]. Intuitively, an increase in a model's depth can be interpreted as providing a model with potentially higher number of consecutive transformations for processing their respective inputs. Similarly, an increase in network width can be interpreted as providing the model with more number of units/features at each processing stage. To examine the effect of these factors on neural prediction generalization, we compared the neural predictivity score across variations of ResNet architecture which differed in their relative depth and width. We considered ResNet architecture with 18, 50, and 101 layers

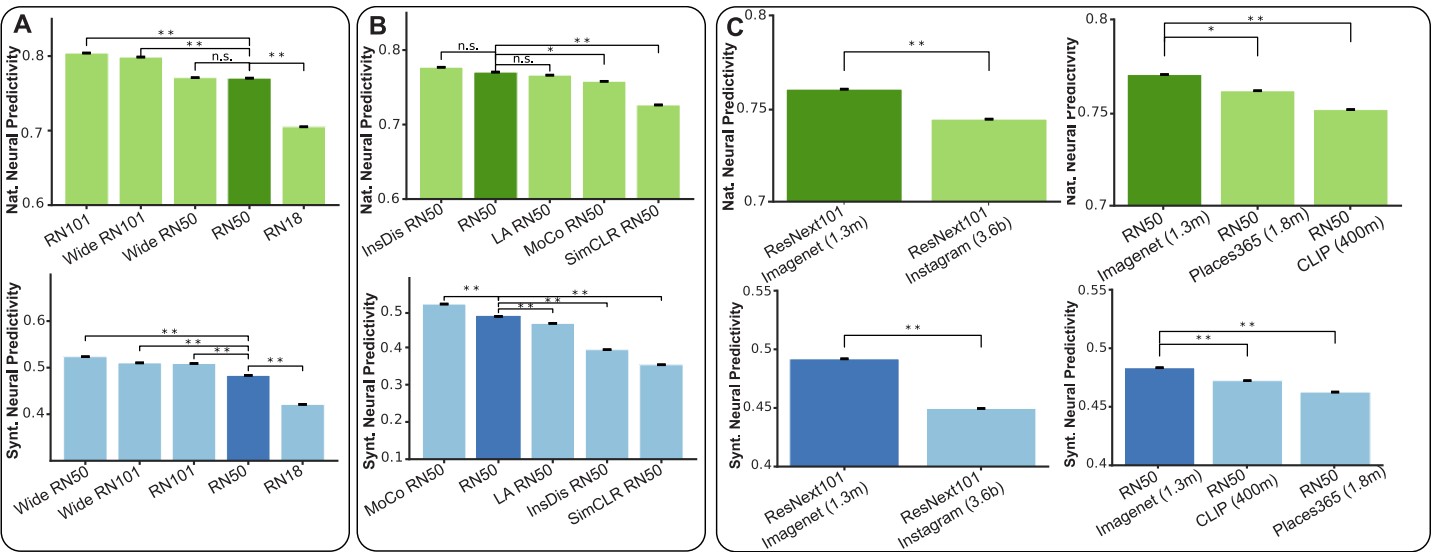

**Fig 6. Design choices affect neural predictivity.** A: Barplot shows the neural predictivity score for different variations of the ResNet architecture on naturalistic (top) and synthetic (bottom) domains. Deeper and wider networks tend to yield higher neural predictivity scores. B: Unsupervised learning algorithms do not significantly improve neural predictivity on natural stimuli while Momentum Contrast leads to significantly higher neural predictivity on synthetic stimuli. C: Larger datasets with substantially higher number of categories (Instagram) and examples (CLIP) do not improve neural predictivity in either stimulus domains. All error bars denote the variance across 5 repetitions of each analysis.

of computation and wide variations of ResNet50 and ResNet101 in which the number of units in each layer are doubled compared to their respective original architectures.

Increasing the depth and the width of the model, both improved the neural prediction accuracy of the model on both naturalistic and synthetic domains (except Wide ResNet101 which did not improve the ID neural predictivity score compared to ResNet101 as well as OOD neural predictivity score compared to Wide ResNet50; Fig 6A). Increasing the depth from 18 to 50 layers had the highest impact on both in- and out-of-distribution neural predictivity scores, but further increasing the depth to 101 had a relatively smaller effect. Likewise, the effect of width was most significant for ResNet50 architecture and to a lesser degree for ResNet101.

**Learning algorithm.**   While much of the initial progress in learning visual representations with neural networks was accomplished through supervised learning, various approaches to unsupervised representation learning have also been developed in recent years [45–48]. Representations learned with these algorithms can perform competitively on neural predictivity measures compared to their supervised-trained counterparts [27].

We additionally investigated the effect of learning algorithm on models' neural predictivity by comparing neural predictivity score across several supervised and unsupervised training approaches while keeping the model architecture fixed. We considered several recent unsupervised learning approaches including SimCLR [45], Momentum Contrast (MoCo) [25], Instance Discrimination [48], and Local Aggregation [47]. We observed significant variations in neural prediction accuracy across different learning algorithms. On the naturalistic domain, none of the unsupervised learning algorithms significantly exceeded the neural predictivity score of the supervised-trained model. In contrast, on the synthetic domain, one such algorithm (MoCo) substantially improved the neural predictivity score beyond the supervised-trained model (Fig 6B).

To verify the generality of this observation, we additionally trained a deeper (ResNet101) and a shallower (ResNet18) neural network with the MoCo algorithm and evaluated their neural predictivity score on both image domains. We found that MoCo similarly improved the neural predictivity score on synthetic images for other variations of ResNet architectures as well (S4 Fig).

**Training dataset.** It has been suggested that models' visual diet is critically important for learning brain-like representations [49, 50]. To investigate this proposal in the context of predictive models of neurons, we tested how training on different datasets affect the neural prediction measures in different stimulus domains. For this, we considered two model architectures (ResNet50 and ResNext) and four datasets including ImageNet (1 million, 1000 categories), CLIP (400 million image and text pairs collected from internet), Place365 (1.8 million, 365 categories), and IG (3.5 billion Instagram images). In both comparisons, we found that the model trained on ImageNet dataset yielded significantly higher neural predictivity on both stimulus domains compared to alternative datasets (Fig 6C). This result suggested that among the other tested alternatives, ImageNet dataset remains as one of the best image datasets for training visual models that can mimic the primate visual representation.

## Do different models predict the neuronal responses the same way?

The results in previous sections demonstrated that different models vary widely in their ability to predict the neural responses in area V4 and these variations are higher when models are validated on out-of-distribution synthetic stimuli compared to naturalistic ones (0.68—0.84 for naturalistic stimuli vs. 0.35—0.61 for synthetic stimuli; see Fig 3D). However, despite the differences, it remains unclear whether predictions made by these models are overlapping or complementary. For example, do different models consistently find the same neuronal sites to be "easy" or "hard" to predict? Do different models consistently predict the same subset of neurons and fail to predict other "difficult" ones? Are there neurons that are consistently not predicted by any of the models? Do models with the same level of predictivity score, predict image responses similarly?

To investigate these questions, we first examined how the neural predictivity scores obtained for different neural sites co-vary across models. For each pair of neuronal models, we computed the vectors containing the predictivity scores obtained for all neural sites for a given domain (naturalistic or synthetic). We then computed the correlation between these vectors for all pairs of models in our pool. We performed this analysis on two levels: 1) *within-model*: where we computed the neural predictivity scores for neuronal models from all layers of the same neural network architecture and then computed the correlations between those scores across pairs of neuronal models from different layers (5 DNN models; Fig 7A; score consistency histogram, top row); 2) *across-models*: where we computed the correlations between pairs of neuronal models each from a different neural network model (38 DNN models; see the full list in section Model selection; Fig 7A; score consistency histogram, bottom row).

Results indicated that models constructed from different layers of the same architecture highly agree with each other on the relative predictivity scores assigned to different neuronal sites. In other words, these models consistently found the same neurons to be easy or hard to predict. Moreover, the consistency was higher for the naturalistic dataset (0.99 ± 0.003) compared to the synthetic one (0.88 ± 0.04). We also found a similar overall pattern in score consistency when considering the cross-model consistencies with slightly lower correlations across models constructed from different architectures (0.94 and 0.76 for naturalistic and synthetic stimuli respectively). The high consistency found across neuronal models stemming from the same neural network model suggests, that across the layers, information about different

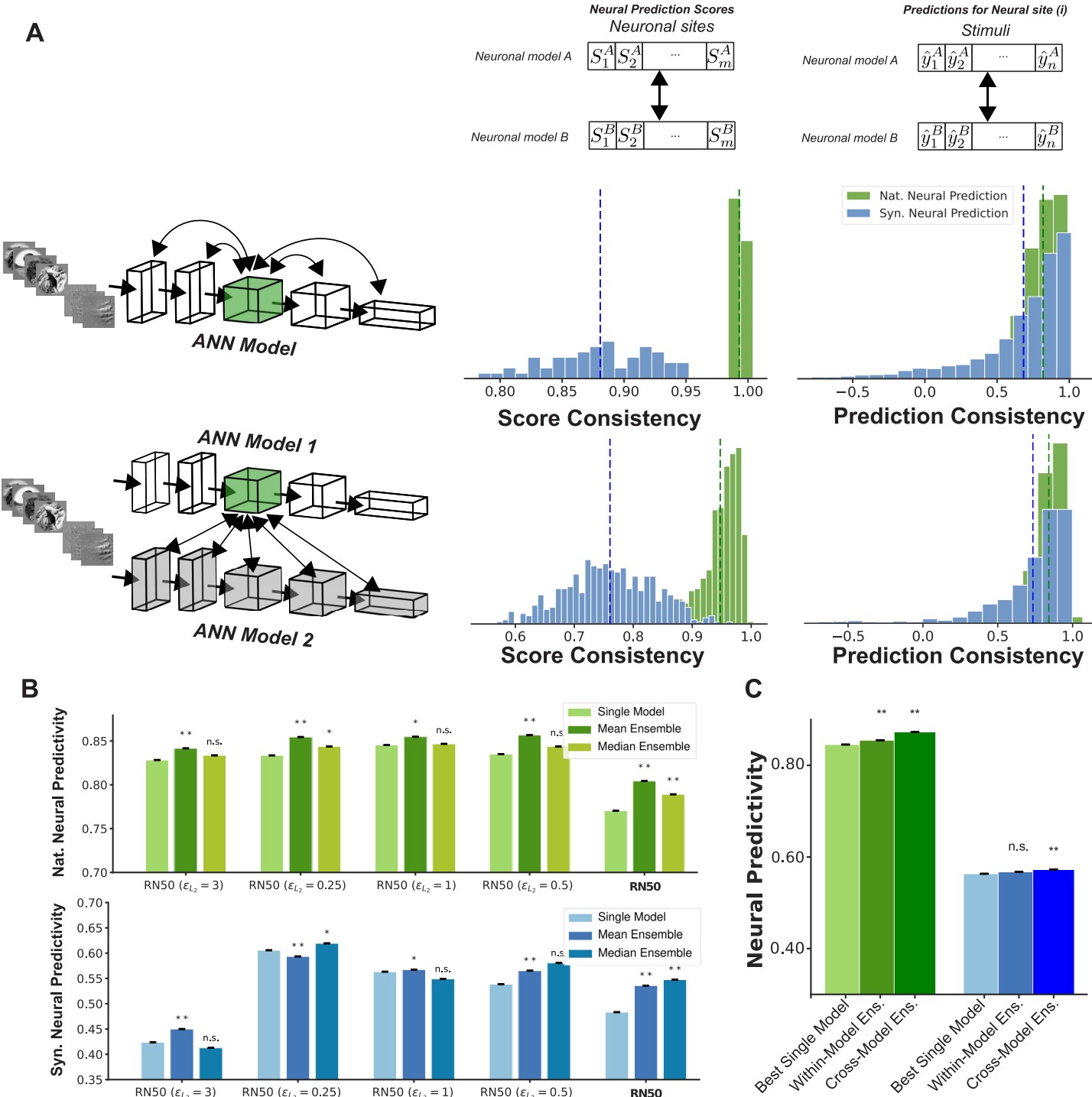

**Fig 7. Prediction and neural prediction score consistency across neuronal models.** A: (left column) histogram of consistency in neuronal predictivity scores of different neuronal sites across neuronal models constructed from different layers within the same neural network architecture (top row) and across layers from different neural network architectures (bottom row). (right column) similar to the left column except the consistency is computed for image-level predictions; B: Comparison of neuronal predictivity scores between single and ensemble neuronal models in natural (top) and synthetic (bottom) domains. The single layer model consists of the best neuronal model for each neuronal site within a neural network model. Each ensemble model is constructed by aggregating the predictions from the top-5 neuronal models within the same network for each neural site; C: comparison of neural predictivity scores in natural and synthetic domains for best single model, within-model and cross-model ensembles. All error bars denote the variance across 5 repetitions of each analysis.

neuronal sites vary at similar rates, while the rank order across neuronal sites remains more or less constant. This means that in most cases, the rank order of the neural predictivity scores across neuronal sites is highly similar for neuronal models constructed from different layers of the same neural network architecture (i.e. neuronal sites with highest and lowest neural predictivity scores in one layer are likely to obtain the highest and lowest neural predictivity scores in other layers as well). This trend is also evident by visually inspecting the graphs in Fig 3A and S3 Fig.

We next asked how consistent are the image-level predictions for neuronal models that predict a neural site with the same level of predictivity score. To investigate this, for each neuronal site, we selected 5 neuronal models with similar predictivity scores ($\leq 2\%$ difference in neural predictivity score) and measured the consistency in image-level predictions across these models by computing the pairwise correlation between vectors of per-image predictions generated by each of the models. Similar to the score consistency analysis, here also we carried out this analysis both within- and across- models. We found that, despite the close similarity in predictivity score for all the considered neuronal models, the distribution of these correlations spans a wide range of values (Fig 7A; prediction consistency histograms). In other words, neuronal models with almost identical predictivity scores, can produce widely different image-level predictions.

Given the apparent inconsistencies in neural predictions across different models, we next asked if combining predictions across models could further enhance the accuracy and generalization of individual neuronal models. We tested this by considering two forms of ensemble predictors: 1) *Within-model ensemble.* where we formed an ensemble predictor by considering the predictors from five layers (within one single model) with highest in-distribution neural predictivity score for each neuronal site; 2) *Cross-model ensemble.* where we formed an ensemble predictor by considering the top 5 layers in the top 5 models for each neural site. To form the ensemble predictor under each setting, we also considered two operators (mean and median) for aggregating the predictions across models.

We found that forming ensembles from layers within the same architecture can improve the neural predictivity score in both naturalistic and synthetic domains. The mean-ensemble predictors consistently improved the model predictivity score on in-distribution samples while median-ensemble predictors more frequently improved the model predictivity score on out-of-distribution samples (Fig 7B). Moreover, cross-model ensemble predictors further improved neural predictivity scores in both in- and out-of-distribution settings beyond single model and within-model ensembles (Fig 7C). On the naturalistic domain, within- and cross-model ensembles improved the neural predictivity score by 1.2 and 3.3 percent respectively above the best single-model. Likewise, on the synthetic domain, this improvement was 0.7 and 1.6 percent.

## Discussion

We investigated the generalization properties of neural network models of visual cortex on their ability to predict the neural responses in macaque's area V4 to out-of-distribution synthetically generated stimuli. By testing a large set of neural network models of visual cortex, we identified a significant gap in neural prediction accuracy between naturalistic and synthetic out-of-distribution samples that was consistent across models. While there are various measures of representational similarity analysis, neural prediction accuracy remains one of the leading measures for evaluating model-brain representational similarity. The neural prediction accuracy is often measured using naturalistic stimulus sets and it's been shown that in some cases the existing models approach the maximum prediction accuracy supported by collected

datasets. This fact has given the impression that the existing models may indeed be getting closer and closer to the brain representations to a degree that they cannot be distinguished from it. Our analyses show that while many of the existing models are indeed approaching the ceiling of neural prediction on naturalistic datasets, the accuracy of these predictions are significantly diminished on OOD samples such as synthetically generated images. These observations are congruent with prior reports in showing lower neural prediction accuracy on synthetically generated stimuli compared to those containing natural scenes and objects [23, 24] and extend these findings to a larger group of neural network models.

## Behavioral and neural generalization

The present work was motivated by the question of whether neural network models with better out-of-distribution object recognition properties are getting us closer to having more general models of the neural responses in the primate visual cortex. Our analyses using a large battery of neural network models from recent computer vision literature and several common object recognition benchmarks suggest a negative answer to this question. Specifically, we found that the standard measures of object recognition generalization from the computer vision literature are not reliable indicators of the models' ability to reproduce the firing response patterns in primate visual cortex. These results question whether further improvements in neural network models' out-of-distribution generalization performance would translate into advances towards more accurate models of neural responses in the animal brains. Although recent work has provided support for the functional role of area V4 in object semantic processing, other computational objectives such as 3D visual representations may have complementary roles in shaping the neural response patterns along the ventral stream including in area V4 [31].

In addition, we found that robustness to adversarial attacks was the only measure which was predictive of a model's neural prediction accuracy on naturalistic images. While robustness was not significantly correlated with neural prediction accuracy on synthetic images, we observed that all of the 5 models with highest OOD neural predictivity (among the 38 tested models) were those that their parameters were tuned to improve adversarial robustness. This finding was strongly suggestive of the positive role of adversarial robustness in constructing models that better align with brain's neurons on the wider range of stimuli. These findings were also inline with recent findings showing closer population characteristics in adversarially robust networks compared to V1 neurons in mice and macaques [51, 52].

## Model-brain alignment

There is recent evidence suggesting that the function performed by a single cortical column may be more complex than that performed by a single typical ANN layer [53]. Popular neural modeling benchmarks such as BrainScore [28] are designed based on the implicit assumption that single layers in neural network models correspond to single areas in the brain and consequently penalize models in which the representation associated with a particular cortical area is spread out across multiple layers. In this work, we argue that this assumption may be over-restrictive and may lead to lower neural predictivity scores in most models of the visual cortex. While relaxing this assumption, we found that neurons in area V4 can be better approximated by units in multiple neural network layers (often 2–3 layers) and importantly that these layers often are not aligned when considering stimuli from different domains (i.e. natural vs. synthetic). These results corroborate prior work which examined the pairwise correlations between DNN units and individual neuronal sites and reported finding DNN units with highest correlation with individual neuronal sites clustered in one of several DNN layers [54].

We also found many neuronal sites in the V4 cortex to be highly predictable by simple representations forming in the early layers of neural network models. However, the neuronal models that stemmed from later layers, in many cases, also generalized better to out-of-distribution stimuli. This suggested that while neural responses may be explained equally well by linear combination of simpler features found in early layers of neural networks, the more complex features are needed to provide an explanation of neural activity that generalizes outside of the natural domain.

Altogether, our results suggest that naturalistic stimuli alone may not be sufficient to assess the similarity of the models of ventral visual cortex to the brain. Synthetically generated stimuli such as those considered in this work or others that are specifically generated for such comparisons (e.g. "controversial stimuli" from [55]) may provide a broader and more effective test bed for model-brain comparisons in the future.

## Effect of architecture and learning rule on generalization

We also observed a trend where deeper models consistently improve the model predictions on naturalistic stimuli which was generally consistent with prior benchmarks of the neural predictivity in macaques [28]. Moreover, we showed that this improvement in prediction accuracy also generalizes to out of distribution synthetic samples.

Prior work which considered only naturalistic samples reported that models trained with unsupervised learning approaches are unable to improve neural predictivity beyond those that are trained supervised [27]. In our analyses, we observed that when model parameters are tuned with a specific unsupervised algorithm (Momentum Contrast), the learned internal representations of these models generalize better to out-of-distribution stimuli compared to their supervised-trained counterparts. This observation re-echos the need for expanding the stimulus sets considered in evaluating models of visual cortex in primates and potentially other animals.

We also compared the model representations when their parameters were trained on different datasets. We showed that while there are large variance in neural predictivity across models trained on different datasets, the moderately sized (with today's standards) ImageNet dataset remains the best dataset for training neural network models with highest neural predictivity. Indeed, we observed that even datasets with orders of magnitude more samples do not yield models with better neural prediction generalization. Our results were also congruent with other recent findings showing that models trained on visual object recognition yield higher brain-similarity than those trained to perform scene recognition [21, 50]. Overall, our results suggest that more data alone may not be sufficient to close the gap in neural predictivity in current models of primate ventral visual cortex.

## Model ensembles

Finally, we showed that different neuronal models constructed from various neural network architectures are highly consistent in their ability to predict the responses across different neuronal sites. This high level of consistency could be indicative of common limitations across existing models of visual cortex that cannot be explained by the visual diet and may be rooted in the architectural motifs or learning algorithms used to tune their parameters. On the other hand, we observed that models with nearly identical predictivity scores exhibit large inconsistencies in their image-level predictions. This encouraged us to combine predictions from multiple models into ensemble predictors which in turn further improved the neural prediction accuracy on both naturalistic and synthetic stimuli. Ensemble models have widely been used in the literature to improve robustness to noise and our results suggest that large ensemble

predictors could further improve the accuracy of these models in predicting neuronal responses to out-of-distribution samples.

## Methods

### Neural dataset

We adopted the publicly available neural recording dataset from [24], which contains measurements of neuronal spiking activity recorded from area V4 of three macaque monkeys (monkey M, monkey N and monkey S) to a set of naturalistic (ID) and synthetically generated images (OOD). The neural firing rate was calculated by counting the number of spikes in a 100 msec window post-stimulus presentation (70–170 msec). Each stimulus was shown to the animal subject multiple times (31–51 times) and the average response for a given neural site across repetitions was used in the analyses.

The naturalistic image set included 640 gray-scale images. Each image contained the rendering of a 3D object instantiated at a random location, size, and pose and overlaid on a random natural image ground. A total of 64 unique object models were used that were grouped into 8 categories with 8 instances of each object category. The eight object categories consisted of animals, boats, cars, chairs, faces, fruits, planes, and tables. The synthetic image set consisted of synthetically generated pixel patterns that were produced using an optimization procedure described in [24]. Briefly, a predictive neuronal model of each neural site was constructed from the internal activity of the AlexNet model [56]. New stimuli were then synthesized that were predicted to 1) produce high activity in individual neuronal sites (stretch) or; 2) produce high activity in an individual neuronal site while suppressing activity in other simultaneously recorded sites (one-hot population). The resulting gray-scale images contained complex curvature and texture like patterns but importantly did not contain any nameable objects.

### Neural predictivity measure

We use prediction accuracy as our primary metric for assessing representational similarity between models and the brain. For each neural network model, we freeze the pretrained (or untrained) neural network weights in all layers. Because of the high-dimensionality of the activations in typical neural network layers, we firsts compute a low dimensional representation for the unit activations in each layer of the neural network model using the following procedure:

1) we compute the model's unit activations at all of its layers in response to 1000 random images from the ImageNet dataset [32]; 2) we apply PCA on each layer's unit activations. Using the set of ImageNet images for computing the PC dimensions allows us to use a single set PCs for all analyses applied on the representation in each layer of each network; 3) we compute the unit activations at each layer of the network to the stimulus set of interest; 4) we use the PCA coefficients derived from step 2 to reduce the dimensionality of the unit activations in response to the stimulus set.

To report the neural predictivity score on the naturalistic data, we consider each neuronal site recorded in each session from each animal as an independent neuronal site. Using a 10-fold cross-validation procedure, we fit Partial Least Squares (PLS) Regression model with 25 components on the train set to predict the neuronal responses in each neuronal site to the held-out test samples from the PC-projected unit activations. For each neuronal site, we compute the Pearson correlation between the predicted and actual responses for each neuronal site. The cross-validation procedure is repeated 5 times for each candidate representation.

We confirmed that naturally-tuned PCA transformation was not a confound in predicting neural responses to synthetic stimuli by performing an additional experiment in which the

generalization gap was measured directly with a PLS regression model and without PCA. The results from this experiment were almost identical to those where PCA and PLS were used together (S8 Fig).

The neural predictivity score for a neural network model is then computed using one of the following two approaches:

1. **Layer-Area mapping.** For each layer, we compute the median correlation across all neuronal sites. We then compute the average of median correlations across the 5 repetitions of the cross-validation procedure and consider that as the neural predictivity score for that layer. We select the layer with the highest predictivity score on the naturalistic stimulus set as the best layer.

2. **Layer-Neuron mapping.** We compute the average prediction accuracy across repetitions for each neuronal site and each layer. For each neuronal site, we select the layer with the highest average cross-validated prediction accuracy on the naturalistic stimulus set as the best layer. We used this approach for the majority of our analyses in the paper.

To report the neural predictivity score on the synthetic data, we compute the neural predictions from the layer with best natural predictive score to the synthetic stimuli using each of the cross-validation models (10) and average those predictions per stimulus across folds. We then compute the Pearson correlation between the averaged predictions and the neuronal responses for each repetition of the cross-validation procedure, take the median across neuronal sites for each repetition and finally, the averaged correlation across repetitions of the cross-validation procedure is reported as the neural predictivity score on synthetic stimuli for that neural network model.

## Generalization gap in neural prediction

Each neuronal model can generate predictions of the neuronal responses to any stimulus that can be formatted as a static image. We assess the generalization ability of each neuronal model by quantifying the gap in the prediction accuracy made on two input domains. In all analyses except those in Fig 2E, we fit the regression parameters to predict the brain responses to naturalistic images (i.e. in-distribution or ID) and use the fitted model to additionally predict the responses the synthetic (out-of-distribution or OOD) stimuli. We then quantify the generalization gap in neural predictivity. For this, we first compute the difference between the predictivity scores on ID and OOD domains for each site on each repetition of the cross-validation procedure. Then, for each repetition, we compute the median difference value across neuronal sites. Finally, we compute the mean of the median values across repetitions. Concretely, we compute the following value:

$$C_{gap} = C_{nat} - C_{synth} \tag{1}$$

where $C_{nat}$, $C_{synth}$, and $C_{gap}$ denote median prediction accuracy (i.e. correlation) on naturalistic imageset, synthetic imageset and the generalization gap in neural predictivity respectively.

## Evaluating object recognition accuracy and object recognition generalization

The ventral visual stream is hypothesized to serve a critical role in visual object recognition. We use object recognition accuracy as a measure of usefulness of the representations in each model of the object recognition behavior. Similar to the approach for assessing neural prediction accuracy, here we assess each model on its ability to recognize objects on different image

datasets. For this, we consider several standard benchmarks of object recognition in computer vision and deep learning.

**In-distribution object recognition accuracy.** We use the ImageNet dataset to assess each model's object recognition accuracy in naturalistic domain. Images in the ImageNet dataset contain iconic views of 1,000 object categories as naturally occurring in the daily life. The ImageNet dataset consists of separate train and validation sets. The train set which consists of ∼1.28 million images is used for parameter tuning in most of the network models considered in this work. The validation set consist of 50,000 images from the same 1,000 object categories and is used to assess the object recognition accuracy of each model on this dataset.

**Out-of-distribution object recognition accuracy.** We consider four standard datasets for evaluating each model's out-of-distribution generalization in object recognition behavior.

- **ImageNet-Rendition.** This dataset contains 3,000 naturally occurring examples which are renditions (e.g., paintings, embroidery, etc.) of 200 object classes in ImageNet classes, but with textures and local image statistics unlike those of ImageNet images. This dataset is designed to assess a model's generalization to various abstract visual renditions, considering that ImageNet-1K primarily contained photos of objects in the wild.

- **ImageNet-Adversarial.** This dataset contains 7,500 natural images, which are selected by an *adversarial* selection procedure to find images that are consistently misclassified by ResNet50 model trained on ImageNet dataset(a popular vision model in machine learning literature). The images are selected from 200 classes from ImageNet dataset to cover a broad range of object categories spanned by ImageNet-1K.

- **ImageNet-Sketch.** This dataset contains 50 hand-drawn sketch-like images for each of 1,000 ImageNet classes (50,000 images in total), which matches the categories and scale on ImageNet validation dataset. The images are collected by Google image queries "sketch of X", where "X" is a class name of ImageNet. The images are within "black and white" color scheme, and with very little or no texture compared with other domains.

- **ImageNet-C.** This dataset is widely used for benchmarking robustness to image corruptions and consists of 15 different corruption types, each at 5 levels of severity, resulting in 75 distinct corruptions. The corruptions can be divided into 4 categories: noise (Gaussian, Shot, Impulse), blur (Defocus, Glass, Motion, Zoom), weather (Snow, Fog, Frost, Bright) and digital (Contrast, Elastic, Pixel, JPEG). ImageNet-C [15] is generated by applying 75 algorithmic corruptions on ImageNet validation dataset.

**Adversarial robustness.** Recent work has revealed that predictions made by deep learning models are extremely sensitive to input changes [34], a phenomenon that is usually referred to as adversarial examples. Adversarial examples are example stimuli that are generated through an optimization procedure that discovers minuscule perturbation patterns that when applied on naturalistic inputs, cause a deep learning model to make large errors in their predictions. Robustness against adversarial attack is of crucial importance in order to make machine learning systems more reliable and potentially more aligned with human judgements. While there are numerous approaches for generating *adversarial examples*, all adversarial attack models involve an optimization procedure through which the pixel values in an input image are adjusted to maximize a measure of model's prediction error. To make sure the perturbed images are indistinguishable from the original inputs, the input perturbations are restricted to remain within a certain bound around data points. The size of the perturbation is usually measured using an $L_p$ norm such as $L_{inf}$ or $L_2$ norm with a certain magnitude $\varepsilon$. We quantified

each model's adversarial robustness as the averaged classification accuracy($A_{adv}$) on adversarially perturbed ImageNet images using PGD-$L_2$ and PGD-$L_{inf}$ with various $\varepsilon$ values.

## Model selection

We considered 38 neural network models in our main analyses. These models inlcude AlexNet, ResNet50, ResNet101, wide ResNet50, wide ResNet50–4, ResNet18, MoCo ResNet50, MoCo ResNet18, MoCo ResNet101, CLIP ResNet50, SimCLR ResNet50, ResNext 101, Weakly Supervised Learning ResNext 101, wide ResNet101, CORnet-S, CORnet-Z, VOneCORnet-S, ResNet50 trained on Stylized ImageNet, ResNet50 trained on Stylized ImageNet and ImageNet, ResNet50 trained on Stylized ImageNet and fine tuned on ImageNet, DeepAugment ResNet50, ResNet50 trained with DeepAugment and AugMix, ResNext50, Adversarial ResNet50 ($\varepsilon_{L_2} \in \{0.01, 0.1, 0.03, 0.5, 0.25, 3, 5, 1, 0.05\}$, step size $= \frac{2 \times \varepsilon}{3}$, Adversarial ResNet50 ($\varepsilon_{L_2} = 3$, step size of $\frac{2.5 \times \varepsilon}{100}$), Adversarial ResNet50 ($\varepsilon_{Li}nf \in \{4/255, 8/255\}$, step size $= \frac{2.5 \times \varepsilon}{100}$), InsDis ResNet50, Local Aggregateion ResNet50, Place365 ResNet50.

**Supervised.** Supervised learning involves training the neural network model on a labelled dataset. In our analyses, we considered models with different architectures (AlexNet, ResNet18, ResNet50, ResNet101, wide ResNet50, wide ResNet101, ResNext50, ResNext101, CORnet-Z, CORnet-S, and VOneCORnet) trained supervised on various datasets (ImageNet, Stylized ImageNet, Instagram-1B) or via various data augmentation (Style Transfer, AugMix and DeepAugment).

- **ResNet family.** Short for Residual Network [2], ResNet is a family of deep neural network models that has been widely used for computer vision tasks. They use residual connections to avoid the vanishing gradient problem that commonly occurs in deep neural networks. ResNet models have different sizes and variations, including ResNet18, ResNet50, ResNet101, wide ResNet50 and wide ResNet101. The number in the model name refers to the number of layers in the model. ResNet50 and ResNet101 are deeper than ResNet18, and have been shown to achieve better performance on many image classification benchmarks. In addition to the standard ResNet models, there are also wide ResNet models, which essentially contain an order of magnitude more filters in each of their layers compared to the original ResNet models.

- **ResNeXt.** ResNeXt models [57] adopt a similar modularized architecture to that in ResNet which relies on stacking convolutional blocks with similar topology to make deeper networks. Unlike in ResNet models, ResNeXt introduces an additional dimension called "cardinality" to improve classification accuracy while maintaining complexity. Moreover, ResNeXt leverages a strategy called "split-transform-merge" (STM), which splits the input into multiple paths, and performs different transformations on the input before the paths are finally merged back together. This allows the network to capture a more diverse set of features from the input data. ResNeXt architecture has been shown to be more accurate than ResNet with fewer parameters, which makes it a more efficient architecture.

**Unsupervised.** Besides supervised trained models, we also considered various unsupervised learning algorithms from the machine learning literature to tune the model parameters. In contrast to supervised learning algorithm, unsupervised learning involves training on unlabelled datasets, with a goal of learning the representation of particular input pattern from image statistics alone. These approaches primarily differ in their learning objectives as listed below.

- **Momentum Contrast.** Momentum Contrast [25] is a self-supervised contrastive learning algorithm for learning visual representations. MoCo consists of a query encoder and a key encoder. The weights of the key encoder are obtained from computing the moving average weights of the query encoder network. The query encoder processes the input data and generates a query vector, while the key encoder processes a large memory bank of previously stored data and generates a set of key vectors. In the context of contrastive learning framework, a similarity score is computed by using the query and key vectors to update the weights of the query encoder.

- **SimCLR.** This method leverages contrastive learning algorithm to learn representations by maximizing agreement between embeddings of two data-augmented views of the same image, while minimizing the agreement to the embeddings of other images [45].

- **Instance Discrimination (InsDis).** This method learns feature representation via instance-level discrimination (as opposed to class-level) [48]. During training, this method treats each image instance as a separate class and trains a classifier to distinguish between each individual image.

- **Local aggregation.** This method train the embedding function to maximize the separation between dissimilar image instances while closing the distance between similar image instances so they are arranged into emerging cluster [47].

Most of the models considered in our analyses were trained using a supervised approach on the ImageNet dataset (i.e. their parameters were optimized to reduce the object recognition error given the ground-truth object labels). However, we additionally considered a number of models from the literature that were trained on alternative datasets, as listed below.

- **CLIP.** Unlike most object recognition models that are trained to predict the object class given an input image, the CLIP model is constructed by jointly training an image encoder and a text encoder by leveraging contrastive training [3]. During this procedure, the network learns to associate the correct caption to each image. At test time, the text encoder embeds the natural language labels to enable zero-shot transfer of the model.

- **Weakly Supervised Learning ResNext 101.** This model is constructed by training a ResNext 101 architecture in a weakly supervised manner on billions of Instagram images to predict the corresponding hashtags as labels [58].

- **ResNet50 trained on Stylized ImageNet, ResNet50 trained on Stylized ImageNet and ImageNet, ResNet50 trained on Stylized ImageNet and fine tuned on ImageNet.** We also adopt 3 models from [59], which are used to study whether CNNs trained on "Stylized ImageNet" can learn a shape-based representation. The models include ResNet50 trained on Stylized ImageNet, ResNet50 jointly trained on Stylized ImageNet and Imagenet, and ResNet50 trained jointly on Stylized ImageNet and ImageNet with fine-tuning on ImageNet.

- **DeepAugment ResNet50.** This models consists of a ResNet50 model trained with DeepAugment approach [60], which is an effective augmentation technique that distorts images by perturbing the internal representation of the deep network.

- **ResNet50 trained with DeepAugment and AugMix.** This model consists of a ResNet50 model trained jointly with DeepAugment and AugMix [60] approaches. AugMix is a data augmentation technique which effectively improves models' robustness by stochastically sampling from a set of augmentation operations to produce diverse augmented images.

- **CORnet models.** CORnet [61] is a family of convolutional neural networks that incorporate recurrent processing into the network architecture. The recurrent loops in this network allow weight sharing across different layers. In our analyses, we considered CORnet-Z and CORnet-S architectures trained supervised on the ImageNet dataset.

- **VOneCORnet-S.** This network is a variation of the CORnet-S architecture where its first layer is replaced with a biologically plausible model of the V1 cortex in primates [52].

- **Adversarially robust models.** We considered several adversarially trained models in our analyses. All models were trained using adversarial training procedures with different norms and epsilons. We used pretrained models from [62] with $L_2$ perturbation of $\varepsilon \in$ {0.01, 0.1, 0.03, 0.5, 0.25, 3, 5, 1, 0.05} using 3 attack steps and a step size of $\frac{2 \times \varepsilon}{3}$. We also adopt publicly available models from [63] with $L_2$ perturbation of $\varepsilon$ of 3 and $L_{inf}$ with perturbation of $\varepsilon$=4/255 and $\varepsilon$=8/255 using 100 attack steps and a step size of $\frac{2.5 \times \varepsilon}{100}$.

## Consistency measure

**Score consistency.** To examine the consistency of different models in predicting the same neurons, we computed the Pearson correlation between the prediction scores made by a pair of neuronal models. We selected the neuronal model under two regimes: 1) when neuronal models were constructed from different layers of the same neural network model; 2) when neuronal models were constructed from layers in different neural network models. For each case, the distribution of consistency (correlation) values were then visualized using a histogram plot. For these analyses, we selected the top 5 layers within each of the top 5 models for this analysis.

**Prediction consistency.** We also evaluated the stimulus-level consistency of different neuronal model predictions. For this, we ran a similar analysis to that for Score consistency where the correlation was computed across the predictions of models for each individual neuronal site. We ran this analysis for random neuronal sites and layer pairs from the top 5 layers within each of the top 5 models.

**Internal consistency.** Neuronal responses often contain inherent variability to the same stimuli. To determine the reliability of the experimental data, we computed a measure of internal consistency for each neuronal site. For this analysis, we analyzed the neural data across different trials where the stimuli were collected (i.e. repetitions). For each neuronal site, we started from a matrix of neuronal responses of size number of repetitions × number of stimuli. We split the matrix into two equal halves by randomly dividing the repetitions into two parts. The average neural response for each half was computed for all stimuli. This resulted in two vectors of the same size as the number of stimuli. We then computed the Pearson correlation between these two vectors. The internal consistency of each neuronal site was computed by applying the Spearman-Brown correction on the correlation ($\frac{2c}{1+c}$), where $c$ represents the correlation between the responses in the two halves. This procedure was repeated 100 times, and the median was taken as the internal consistency of that neuron. In our analyses, we only considered neuronal sites with an internal consistency greater than 0.7.

## Supporting information

**S1 Fig. Scatter plot of neural prediction generalization gap as a function of out-of- distribution object recognition accuracy.** OOD object recognition accuracy for each model is computed as the average accuracy of that model across 5 OOD object recognition benchmarks.

Each dot corresponds to one neural network model. Neural prediction generalization gap is computed as the difference between the neural predictivity on the natural and synthetic domains.
(PDF)

**S2 Fig. Scatter plot of neural prediction accuracy across natural and synthetic domains.** Significant correlation exists between the neural predictivity in natural and synthetic domains. Each dot corresponds to one neural network model.
(PDF)

**S3 Fig. Comparison of neural predictivity score across layers of ResNet50 model for individual neuronal sites.** (left column) Neural predictivity scores from unit activity in each layer of ResNet50 architecture for individual neuronal sites recorded during different sessions and different animal subjects. From top, rows correspond to M-S2, M-S3,M-S4, and S-S2. Colors correspond to the neural predictivity score on natural (green) and synthetic (blue) domains. Different shades correspond to different neuronal site in the same animal. Bold lines correspond to the average predictivity score in each domain across all neuronal sites within that animal's session; (right column) Number of neurons with highest neural predictivity in a given layer corresponding to the same subplot in a. Colors are the same as those in the left column.
(PDF)

**S4 Fig. Momentum contrast improves OOD neural predictivity score on other variations of ResNet architecture.** Comparison of ID (green), OOD (blue), and generalization gap (purple) in neural predictivity on ResNet18 (left) and ResNet101 (right) variations of the ResNet architecture. MoCo improves OOD neural predictivity on both architectures.
(PDF)

**S5 Fig. Scatter plot of natural and synthetic neural predictivity as a function of individual robustness measures.** Top and bottom rows illustrate the scatter plots for natural and synthetic domains respectively. Each column corresponds to one measure of robustness used to compute robust accuracy. Neural predictivity is significantly correlated with robustness only for small values of $\varepsilon$ for each norm.
(PDF)

**S6 Fig. Neural predictivity during training.** Neural predictivity in both natural and synthetic domains increases during training of ResNet50 and ResNet101 neural networks.
(PDF)

**S7 Fig. Histogram of Layer-Neuron improvement over Layer-Area approach for each neuron.** The histograms shows the distribution of difference of ID (left column; green) and OOD (right column; blue) neural predictivity between Layer-Neuron and Layer-Area mapping approach for each neuronal site. The models includes those in Fig 7B. The dashed vertical lines denote the mean of the distribution. A consistent positive mean value illustrates that more than a half of the neurons get their prediction performance improved by using the Layer-Neuron approach.
(PDF)

**S8 Fig. Neural predictivity comparison between training on natural and synthetic data without using PCA.** Comparison of Nat. and Syn. predictivity scores for ResNet50 model when the regression model was fitted on naturalistic data (left) and synthetic data (right) without using PCA. The regression model fitted to the synthetic domain shows worse

generalization to the naturalistic domain similar to Fig 2E.
(PDF)

## Acknowledgments

All analyses were executed using computational services provided by the Digital Research Alliance of Canada (Compute Canada; RRG application 3698). We would like to thank Dr. Daniel Yamins and their group for providing access to the code and checkpoints of the Local Aggregation model.

## Author Contributions

**Conceptualization:** Pouya Bashivan.

**Formal analysis:** Yifei Ren.

**Funding acquisition:** Pouya Bashivan.

**Investigation:** Yifei Ren.

**Methodology:** Yifei Ren.

**Software:** Yifei Ren.

**Supervision:** Pouya Bashivan.

**Validation:** Pouya Bashivan.

**Visualization:** Yifei Ren.

**Writing – original draft:** Yifei Ren, Pouya Bashivan.

**Writing – review & editing:** Yifei Ren, Pouya Bashivan.

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
