## [Decision Letter · Decision Letter 0]

31 Aug 2023

Dear Dr. Bashivan,

Thank you very much for submitting your manuscript "How well do models of visual cortex generalize to out of distribution samples?" for consideration at PLOS Computational Biology.

As with all papers reviewed by the journal, your manuscript was reviewed by members of the editorial board and by several independent reviewers. In light of the reviews (below this email), we would like to invite the resubmission of a significantly-revised version that takes into account the reviewers' comments.

Please address R2's concerns over readability and whether the use of PCs of layer activations is suitable when generalising the model from natural to synthetic data.  

We cannot make any decision about publication until we have seen the revised manuscript and your response to the reviewers' comments. Your revised manuscript is also likely to be sent to reviewers for further evaluation.

Sincerely,

Emma Claire Robinson

Academic Editor

PLOS Computational Biology

Marieke van Vugt

Section Editor

PLOS Computational Biology

Reviewer's Responses to Questions

**Comments to the Authors:**

Reviewer #1: The authors show a study of comparing similarity between a trained artificial neural network with the responses recorded from V4 neurons in the monkey brain. They find that while network accuracy is correlated with the predictability of the neurons responses for the natural images, it is not the case for synthetic images. The authors show this for a range of different networks.

For similar ImageNet and similar dataset that are aimed to confuse the network (adversarial, sketches, corruptions) they do not see a correlation between performance and accuracy) but for training on adversarial images they see an improvement of neural predictivity for networks trained to be robust to adversarial attacks.

The analysis seems well done and covers a broad spectrum of scenarios where they show that while current networks are good at prediction neural activity from natural images, they fail to predict the brains response to synthetic images.

I found some minor comments on their article:

Figure 2 caption. The caption does not describe subfigure d and e

Line 146: There is no Fig 1e. I assume the authors are referring to Fig 2e

Fig 3 b. Why is there only a very small number of neurons explained (now many are there in total?) and why are they clustered to specific layers?

Fig 3 c. The Distributions in 3b suggest that the histogram of the number of neurons is shifted left in the case of the green line (natural images) compared to the blue line (synthetic images), but Fig3c shows the opposite. Are the histograms in 3b atypical?

Line 200: You say that on out-of-distribution samples the Layer-Neruon mapping improved substantially, yet the 0.487 and 0.479 are very close, also given the big variety in the data Fig 3d and a considerable number in Fig 3d also decreases from LA to LN.

Fig 4b: the bars are in a different order than the legend. Please order the legend the same as the bars to avoid confusion. Also in the text the order in which the alterations are explained is switched between ImageNet Adversarial and ImageNet Rendition, adjusting the order to match Fig 4a might be good.

Fig 5c: you omitt two of the values from 5b, the 4/255 and 8/255, why?

Reviewer #2: This research sought to answer the pivotal question of whether the OOD generalization capacity in task-performing DNNs mirrors the OOD generalization in neural prediction. The authors initially demonstrated that state-of-the-art neural predictive models fail to generalize on OOD synthetic stimuli. They then explored whether more adversarially robust models better predict neural activity. The study is interesting. However, I think the overall structure of the article feels somewhat disjointed. It would be more reader-friendly if each section were organized around a clear theme, specifically outlining why that section is essential to answering the primary question. Major revisions are required before publication. Please see below.

I have a question about the predictivity measure (lines 552-558). Since the authors utilized PCs of layer activations to reduce dimensionality, were the PCs estimated from natural images also used to train the predictive model on synthetic data? If this is the case, isn't it unfair for predicting OOD responses? Many pertinent features might exist in the original model layer but might be omitted due to the projection to PCs from another domain. I'm curious why the authors didn't simply regress the raw model features to neural responses. Considering it's merely a linear mapping, I assume it would still be feasible.

Monkey S's data exhibit similar predictivity to both natural and synthetic stimuli in both sessions (Fig 2c). It would be insightful to discuss how this monkey's data differs from others.

Section 2.2 contrasts Layer-Area mapping with Layer-Neuron mapping. As indicated by Fig 3a and 3d, the disparity isn't significant. I feel this section doesn't critically support the main argument. It might be more appropriate to directly use the layer-neuron analysis and relegate the comparative discussion to supplementary materials.

The title of section 2.3 might be more aptly named "Is Recognition Generalization Ability a Reliable Indicator of Neural Predictivity?".

A comprehensive list of models employed in the analysis should be included. For instance, line 113 mentions "40 artificial neural networks", and line 216 refers to "33 neural network models". Which are these models? Were they developed by the authors, or sourced from prior studies? If the former, will they be made publicly available?

The figures are meticulously crafted and lucidly depicted; I commend the authors for their dedication.

Minor points:

• On line 146, "Fig 1e" should be corrected to "Fig 2e".

• Captions for Fig 2d and 2e appear to be absent.

• In Fig 3, incorporating a legend for "green" and "blue" would enhance clarity.

• The bibliography section needs revision; numerous references lack titles.

• There are numerous error bars in the bar plots, like in Fig 2c, 2d, Fig 5b, Fig 6, etc. How are these defined? Are they standard deviations or standard errors of the mean? Furthermore, over which variables are these calculations made?

**Have the authors made all data and (if applicable) computational code underlying the findings in their manuscript fully available?**

Reviewer #1: **No: **While they use publicly available data, the code is not (yet) available, the authors write that they will do so upon publication.

Reviewer #2: None

PLOS authors have the option to publish the peer review history of their article (what does this mean?). If published, this will include your full peer review and any attached files.

Reviewer #1: No

Reviewer #2: No
---

## [Decision Letter · Decision Letter 1]

29 Apr 2024

Dear Dr. Bashivan,

We are pleased to inform you that your manuscript 'How well do models of visual cortex generalize to out of distribution samples?' has been provisionally accepted for publication in PLOS Computational Biology.

Best regards,

Emma Claire Robinson

Academic Editor

PLOS Computational Biology

Marieke van Vugt

Section Editor

PLOS Computational Biology

Reviewer's Responses to Questions

**Comments to the Authors:**

Reviewer #1: The authors have addressed all my previous concerns.

**Have the authors made all data and (if applicable) computational code underlying the findings in their manuscript fully available?**

Reviewer #1: **No: **The authors provide the code, but the code for the figures loads the data from their private google drive. I did not find a link to their data.

PLOS authors have the option to publish the peer review history of their article (what does this mean?). If published, this will include your full peer review and any attached files.

Reviewer #1: No

---

## [Editor Report · Acceptance letter]

24 May 2024

PCOMPBIOL-D-23-00694R1 

How well do models of visual cortex generalize to out of distribution samples?

Dear Dr Bashivan,

I am pleased to inform you that your manuscript has been formally accepted for publication in PLOS Computational Biology. Your manuscript is now with our production department and you will be notified of the publication date in due course.

With kind regards,

Olena Szabo
